# Association between Periodontal Condition and Fat Distribution in Japanese Adults: A Cross-Sectional Study Using Check-Up Data

**DOI:** 10.3390/ijerph20031699

**Published:** 2023-01-17

**Authors:** Koichiro Irie, Tatsuo Yamamoto, Tetsuji Azuma, Komei Iwai, Takatoshi Yonenaga, Takaaki Tomofuji

**Affiliations:** 1Department of Dental Sociology, Kanagawa Dental University, Yokosuka 238-8580, Japan; 2Department of Community Oral Health, School of Dentistry, Asahi University, Mizuho 501-0296, Japan

**Keywords:** periodontal condition, fat distribution, Japan

## Abstract

Some studies have reported a significant association between periodontal condition and obesity status. We hypothesized that visceral fat area (VFA) and subcutaneous fat area (SFA) volumes might be correlated with periodontal condition. The purpose of the present cross-sectional study was to investigate the association between periodontal condition and fat distribution in Japanese adults. A total of 158 participants, aged 35–74 years, underwent a health check-up including fat distribution and oral examinations. All of the participants underwent magnetic resonance imaging to quantify VFA and SFA. Periodontal condition was evaluated using the periodontal pocket depth (PPD) and clinical attachment level. The VFA volumes differed among the PPD score and clinical attachment level (CAL) code groups. On the other hand, no significant differences in SFA volume were observed among different periodontal conditions. Multiple regression analysis showed that VFA was positively correlated with a greater CAL (standardized coefficient β = 0.123, *p* = 0.009), but not with a greater PPD score. A larger VFA was positively associated with a greater CAL in Japanese adults, whereas no association was found between SFA and periodontal condition.

## 1. Introduction

Periodontal disease is one of the most common chronic inflammatory diseases and the primary cause of tooth loss. Periodontal diseases caused by oral bacteria induce the production of inflammatory mediators in the systemic circulation [1,2]. Numerous studies have focused on the relationship between periodontal disease and systemic disorders, such as diabetes mellitus, rheumatoid arthritis, cardiovascular disease, and obesity [3,4]. 

The association between periodontal disease and obesity is the most recent area of research in periodontal medicine. Several cross-sectional studies have found significant positive associations between periodontal disease and obesity status, which has also been reported in meta-analyses [5,6]. This evidence indicates that obesity is a risk factor for periodontal diseases. On the other hand, epidemiological studies have also suggested that periodontal disease may be a risk factor for obesity. For instance, it has been reported that subjects with periodontal pockets ≥4 mm showed a significantly higher odds ratio (OR) for onset of obesity at 4 years than those without periodontal pockets [OR: 1.59, 95% CI (confidence interval): 1.08–2.35, *p* < 0.05] [7]. 

Obesity is defined as abnormal or excessive fat accumulation and is usually defined by body mass index (BMI) [8]. However, because BMI does not take into account both frame size and body composition (whether the weight is because of fat or muscle) [9], recent studies have suggested that fat distribution may be more important for predicting the development of several diseases than BMI [10]. Excess accumulation of visceral fat itself increases the risk of atherosclerosis, irrespective of BMI [11]. In addition, compared to subcutaneous fat, visceral fat accumulation significantly increases the risk of type 2 diabetes [12], cardiovascular disease [13], cancer [14], and all-cause mortality [15]. Visceral fat accumulation may affect the progression of several chronic diseases through the production of various inflammatory mediators [16]. These observations suggest that fat distribution is more important than BMI in terms of the risk of many chronic health conditions. Therefore, in examining the association between obesity and periodontal disease, it is important to understand the association with fat distribution as well as BMI. However, only a few studies have investigated the association between periodontal condition and fat distribution [17]. 

Obesity and obesity-related diseases such as diabetes mellitus (DM) and hypertension are major causes of disease burden and increasing health expenditures. The biological link between DM control and periodontal disease has been investigated. Several reviews have been conducted to assess the evidence that periodontal treatment influences DM control [18,19]. Therefore, by clarifying the association between periodontal disease and obesity, if treatment of periodontal disease has the potential to improve obesity, new policy recommendations can be made to reduce healthcare expenditures. 

Visceral fat area (VFA) and subcutaneous fat area (SFA) are commonly used to analyze fat distribution [20]. In the present work, we hypothesized that the volumes of VFA and SFA might be correlated with periodontal condition. As our health check-ups include dental check-ups and the measurement of fat distribution, it is convenient to match periodontal status with VFA and SFA data. The purpose of this cross-sectional study was to investigate the relationship between periodontal condition and VFA and SFA volumes among Japanese adults who underwent health check-ups at the Asahi University Hospital Human Health Center.

## 2. Materials and Methods

### 2.1. Study Participants

This study was conducted in accordance with the Declaration of Helsinki, and the study protocol was approved by the Ethics Committee of Asahi University (No. 30018). A total of 158 participants aged 35–74 years who underwent health check-ups were recruited from January 2016 to December 2018 at the Asahi University Hospital in Gifu, Japan. All of the participants provided written informed consent before enrollment in the study. 

### 2.2. Sample Size Calculation

A prior power analysis was performed with an alpha value of 0.05 and a power of 0.80. According to our previous report [21], the effect size was calculated to be 0.49. Based on our calculations, 53 participants were included in the study. 

### 2.3. Assessment of Body Composition and Fat Distribution

Weight and height for each participant were measured using a body composition meter (Tanita, Tokyo, Japan), and their BMI was calculated. In addition, their abdominal fat composition (VFA and SFA) was determined by the standard method using automated computed tomography (FUJIFILM, Tokyo, Japan) with a special program for fat area measurement [18].

### 2.4. Measurement of Biochemical Markers

Venous fasting blood samples were collected, and fasting blood glucose was determined using an automatic analyzer (Dimension Vista 1500, Siemens Healthineers Japan, Tokyo, Japan; DM-JACK, Kyowa Medex, Tokyo, Japan).

### 2.5. Oral Examination

Three dentists examined the oral health status of the participants. The number of teeth present was then counted. Ten teeth were selected for periodontal examination: two molars in each posterior sextant, and the upper right and lower left central incisors. Periodontal pocket depth (PPD) and clinical attachment level (CAL) [5] were assessed using a periodontal probe (YDM, Tokyo, Japan) at six sites (mesiobuccal, mid-buccal, distobuccal, mid-lingual, mesiolingual, and distolingual) per tooth. Individual PPD values were categorized as follows: score 0 (≤3 mm), score 1 (4–5 mm), and score 2 (≥6 mm). Individual CAL values were categorized into codes 0 (≤3 mm), 1 (4–5 mm), 2 (6–8 mm), 3 (9–11 mm), and 4 (≥12 mm) (CAL code) [22]. The presence or absence of bleeding on probing (BOP) was also recorded. Good intra- and inter-examiner agreement was achieved for repeated PPD and CAL measurements (kappa statistic > 0.8). 

### 2.6. Questionnaire

Lifestyle information was obtained using a self-administered questionnaire [18]. The questionnaire contained three categories of responses (not every day, occasionally, or every day) to describe the participants’ frequency of alcohol consumption and consumption of fish, vegetables, and fruit. The categories for each type of consumption were combined into two categories: not every day and every day. This questionnaire also included items on sex, age, smoking habits (yes or no), and regular exercise habits (presence or absence). 

### 2.7. Statistical Analysis

The measurements and calculated values are presented as medians and interquartile ranges. The chi-square test or Kruskal–Wallis test with a post-hoc Mann–Whitney U test (corrected Bonferroni method) was used to compare the groups with different PPD scores or CAL codes. Spearman’s rank correlation analysis was used to assess the association between VFA, SFA, and clinical variables. Independent variables for the following multiple linear regression analyses were selected for Spearman’s rank correlation analysis when the *p* value was <0.20 for the Spearman’s rank correlation analysis. Multiple linear regression analyses were performed to identify the independent predictors of VFA or SFA. Age, sex, smoking status, dietary behavior, and regular exercise habits were selected as covariates. As there was a strong correlation between PPD scores and CAL codes, they were analyzed separately to avoid multicollinearity. The level of statistical significance was set at *p* < 0.05. SPSS version 25 (IBM Japan, Tokyo, Japan) was used to analyze all data.

## 3. Results

Table 1 presents participants’ characteristics. Approximately 70% of the participants were male. The median (25 and 75 percentile) age, BMI, VFA, and SFA of all of the participants were 54.5 (44.0, 62.0) years, 23.5 (21.75, 26.1) kg/m^2^, 104.5 (61.3, 162.7) cm^2^, and 123.4 (94.3, 176.9) cm^2^, respectively. The PPD of scores ≥1 group were 65.8% and the CAL code of ≥1 group were 76.6%. 

Table 2 shows the differences in VFA and SFA according to periodontal condition. There were statistically significant differences in VFA volumes between PPD scores of 0 and 1 (*p* < 0.001) and between those with CAL codes 0 and 1 or 2 (*p* < 0.001). On the other hand, no significant differences in SFA volume were observed among the different PPD scores and CAL codes.

Table 3 shows the results of the correlation analysis between VFA, SFA, and the other variables. VFA values were positively correlated with sex (r = 0.227, *p* = 0.004), smoking habits (r = 0.390, *p* < 0.001), BMI (r = 0.854, *p* < 0.001), fasting blood glucose (r = 0.437, *p* < 0.001), PPD score (r = 0.251, *p* = 0.001), and CAL code (r = 0.353, *p* < 0.001) and negatively correlated with the presence of regular exercise habits (r = −0.267, *p* = 0.001), frequency of vegetable consumption (r = −0.166, *p* = 0.037), and present number of teeth (r = −0.265, *p* = 0.001). SFA values were positively correlated with smoking habits (r = 0.158, *p* = 0.046), BMI (r = 0.705, *p* < 0.001), and fasting blood glucose (r = 0.161, *p* = 0.043) and negatively correlated with the presence of regular exercise habits (r = −0.309, *p* < 0.001). There were no significant correlations between SFA and PPD scores or CAL codes.

Table 4 shows the results of the multivariate regression analysis with VFA as the dependent variable. When adjusting Model 1, the VFA values were significantly related to BMI (standard β = 0.757, *p* < 0.001), and fasting blood glucose (standard β = 0.113, *p* = 0.015) but not with PPD. On the other hand, when adjusting Model 2, the VFA values were significantly related to BMI (standard β = 0.750, *p* < 0.001), fasting blood glucose (standard β = 0.097, *p* = 0.035), and CAL code (standard β = 0.123, *p* = 0.009). 

Table 5 shows results of the multivariate regression analysis with SFA as the dependent variable. When adjusting Model 1, the SFA values were significantly related to BMI (standard β = 0.705, *p* < 0.001) and fasting blood glucose (standard β = −0.142, *p* = 0.018). On the other hand, when adjusting Model 2, the SFA values were significantly related to regular exercise habits (standard β = −0.142, *p* = 0.019), BMI (standard β = 0.710, *p* < 0.001), and fasting blood glucose (standard β = −0.041, *p* = 0.028).

## 4. Discussion

In this cross-sectional study, we investigated whether periodontal conditions were correlated with fat distribution in Japanese adults. The VFA volume was larger with a higher CAL code. Furthermore, in the multiple regression analyses, the VFA values were positively correlated with CAL after adjusting for sex, smoking habits, regular exercise habits, frequency of vegetable consumption, BMI, fasting blood glucose, present teeth, and BOP. The results indicated that a greater CAL code was positively associated with a larger VFA in Japanese adults. 

However, although the VFA values were higher with higher PPD scores, no significant correlations between VFA and PPD scores were found in the multiple regression analyses. PPD has been reported to be an indicator of current periodontal disease status, whereas CAL is an indicator of cumulative tissue destruction, including past periodontal disease [23]. Therefore, the amount of visceral fat is likely to be influenced by the cumulative progression of periodontal disease rather than the presence or absence of current periodontal disease. 

A possible mechanism by which periodontal condition is related to VFA size is that periodontal disease increases the production of crevicular fluid and induces chemotaxis of polymorphonuclear leukocytes, which release reactive oxygen species (ROS) and inflammatory cytokines into the systemic circulation [24,25,26]. ROS are regulators of mitochondrial activity, alter the concentration of molecules involved in inflammation related to the number and size of adipocytes, and promote adipogenesis and lipogenesis [27]. In addition, other studies have suggested that chronic periodontal pathogens increase cytokine production, thereby increasing the risk of causing type 2 diabetes and cardiovascular complications [28,29]. In in vivo studies, systemic low-grade inflammation after experimental periodontal disease was associated with increased gene expression for adipose tissue levels of interleukin (IL)-6 and C-reactive protein (CRP) in a rat model [30]. Additionally, stimulation of adipocytes with CRP and inflammatory cytokines leads to a greater differentiation of adipocytes [31]. These results indicate that increased systemic levels of ROS and inflammatory cytokines due to periodontal disease could directly or indirectly affect VFA size. 

The link between periodontal condition and fat distribution remains controversial. Our finding might be consistent with a previous study that showed a relationship between CPI codes and VFA in Koreans [17]. However, another report suggested that periodontal disease was not significantly correlated with VFA in patients with type 2 diabetes [6]. A recent study demonstrated that periodontal disease may modulate obesity in female patients [32]. Periodontal diseases can affect overall health by altering the levels of adipokines (IL-1β, leptin, resistin, and adiponectin) in the serum, saliva, and GCF of obese female patients. Collectively, large and longitudinal studies with possible interventions are needed to elucidate the epidemiological link between periodontal conditions and fat distribution and whether there is a bidirectional association independent of other confounding risk factors. However, there was no statistically significant association between periodontal condition and SFA. These results suggest that subcutaneous fat accumulation is not associated with periodontal condition. Visceral and subcutaneous adipocytes may have different properties for the production of bioactive molecules [33]. Some studies have investigated the biological differences between VFA and SFA. The rate of lipolysis and lipogenesis activities is higher in adipocytes of visceral fat tissue than in subcutaneous fat tissue [24,34]. VFA compared with SFA showed elevated gene expression of cytokines, cell adhesion molecules, and key regulators of metabolic homeostasis [35,36]. The unique role of VFA as a highly active endocrine organ may constitute a crucial modifier of obesity. Taken together, VFA might be more sensitive to external factors than SFA. 

Dietary habits are major determinants of obesity [37]. Our results also showed that VFA and SFA were negatively not significant, but they were closely correlated with the frequency of consumption of vegetables. This finding suggests that daily vegetable intake contributes to a reduction in the risk of obesity. 

Furthermore, VFA and SFA has been shown to be positively correlated with glycosylated hemoglobin (HbA1c) in diabetic and non-diabetic populations [38,39]. The adverse influence of visceral adiposity on glucose and lipid metabolism could contribute to the positive correlations between VFA and insulin resistance [40]. This study also showed that VFA values were positively correlated with fasting blood glucose levels. Therefore, fat distribution is an important determinant of insulin resistance and secretion [41].

Regarding the interpretation of our results, the proportion of those with periodontal disease in the analyzed population was 65.8%, which was higher than the proportion of 50.4% among aged 35–74 years from the Japanese Ministry of Health, Labor, and Welfare`s Survey of Dental Diseases in 2016. Furthermore, the average of those with the VFA was 104.5% in the analyzed population, which was higher than the average of approximately 98.0% and 81.6% in other Japanese studies [42,43]. Further research is needed to verify the generalizability of these results. 

The present study had some limitations. First, it had a cross-sectional design, which prevented conclusions regarding causal relationships. Second, the correlation coefficient between VFA and CAL was weak, even though it was statistically significant. This is likely to be because BMI has a strong correlation with VFA and SFA. In addition, we investigated the correlation between BMI and PPD and CAL (data not shown). The rank correlation coefficients of VFA for PPD (r = 0.251, *p* = 0.001) and CAL (r = 0.353, *p* < 0.001) were higher compared to those of BMI for PPD (r = 0.197, *p* = 0.013) and CAL (r = 0.227, *p* = 0.004), respectively. These results are in agreement with previous studies [10]. Third, all of the participants underwent health check-ups at Asahi University Hospital. The participants were regional and were recruited from a single hospital, which might have resulted in overestimation or underestimation due to sample bias. Fifth, since the data were obtained from health check-ups, biomarkers in relation to fat content were not used for the analyses in the present study. 

## 5. Conclusions

In Japanese adults who underwent health check-ups at Asahi University Hospital Human Health Center, there was a positive correlation between increased VFA and higher CAL codes, but no correlation between SFA and CAL codes. In the future, it is necessary to elucidate how visceral fact affects periodontal disease. 

## Figures and Tables

**Table 1 ijerph-20-01699-t001:** Characterization of the study participants (*n* = 158).

Variables	
Male, *n* (%)	107 (67.7)
Age, years	54.5 (44, 62)
Drinking habits (every day)	35 (22.2)
Smoking habits (presence), *n* (%)	29 (18.4)
Regular exercise habits, *n* (%)	30 (19.2)
Frequency of fish consumption (every day), *n* (%)	19 (12.0)
Frequency of vegetable consumption (every day), *n* (%)	114 (72.2)
Frequency of fruit consumption (every day), *n* (%)	57 (36.1)
BMI, kg/m^2^	23.5 (21.7, 26.1)
VFA (cm^2^)	104.5 (61.3, 162.7)
SFA (cm^2^)	123.4 (94.3, 176.9)
Fasting blood glucose (g/L)	103 (95, 109)
Number of teeth present	28 (26, 28)
BOP presence, *n* (%)	36 (22.8)
PPD score, *n* (%)	
0	54 (34.2)
1	82 (51.9)
2	22 (13.9)
CAL code, *n* (%)	
0	37 (23.4)
1	99 (62.7)
2	18 (11.4)
3	4 (2.5)
4	0 (0.0)

Continuous variables are expressed as median (first quartile and third quartile) deviation. BMI; body mass index, VFA; visceral fat area, SFA; subcutaneous fat area, BOP; bleeding on probing, PPD; probing pocket depth, CAL; clinical attachment level.

**Table 2 ijerph-20-01699-t002:** Differences in VFA and SFA condition according to periodontal condition.

	Probing Depth Score			Clinical Attachment Level Code		
	0 (*n* = 54)	1 (*n* = 82)	2 (*n* = 22)	0 (*n* = 37)	1 (*n* = 99)	2 or 3 (*n* = 22)
VFA (cm^2^)	73.8 (41.7, 121.6)	122.2 (73.5, 178.8) *	116.2 (78.6, 168.8)	59.7 (33.3, 99.4)	111.5 (73.6, 177.5) ^†^	150.6 (85.8, 199.1) ^†^
SFA (cm^2^)	118.1 (85.4, 163.1)	123.4 (98.7, 182.1)	141.1 (112.3, 229.0)	117.5 (87.2, 160.1)	122.6 (94.4, 194.4)	133.2 (101.7, 195.2)

Continuous variables are expressed as median (first quartile and third quartile) deviation. * *p* < 0.001, compared with the participants with probing depth score 0, using the Kruskal–Wallis test with the post-hoc Mann–Whitney U test (corrected by the Bonferroni`s method). ^†^
*p* < 0.001, compared with the participants with clinical attachment level code 0, using the Kruskal–Wallis test with the post-hoc Mann–Whitney U test (corrected by the Bonferroni`s method) VFA; visceral fat area, SFA: subcutaneous fat area.

**Table 3 ijerph-20-01699-t003:** Spearman’s rank correlation analysis of VFA and SFA.

Variables	VFA		SFA	
	r	*p* Value	r	*p* Value
Gender (male)	0.227	0.004	0.097	0.224
Age, years	0.099	0.216	−0.049	0.541
Drinking habits (every day)	−0.008	0.919	0.034	0.671
Smoking habits (presence)	0.390	<0.001	0.158	0.046
Regular exercise habits (presence)	−0.267	0.001	−0.309	<0.001
Frequency of fish consumption (every day)	0.020	0.805	−0.013	0.871
Frequency of vegetable consumption (every day)	−0.166	0.037	−0.092	0.249
Frequency of fruit consumption (every day)	−0.114	0.155	−0.137	0.087
BMI (kg/m^2^)	0.854	<0.001	0.705	<0.001
Fasting blood glucose (g/L)	0.437	<0.001	0.161	0.043
Present teeth (*n*)	−0.265	0.001	−0.143	0.073
BOP (presence)	0.144	0.071	0.094	0.242
PPD score	0.251	0.001	0.156	0.050
CAL code	0.353	<0.001	0.114	0.155

VFA; visceral fat area, SFA; subcutaneous fat area, BMI; body mass index, BOP; bleeding on probing, PPD: probing pocket depth, CAL: clinical attachment level.

**Table 4 ijerph-20-01699-t004:** Factors associated with VFA in the study population by multiple regression analysis.

Dependent Variables	Unstandardized Coefficients (B)	95% CI for B		Standardized Coefficients β	*p* Value
		Lower Bound	Upper Bound		
Model 1					
Gender (male)	2.980	−9.565	15.526	0.021	0.639
Age, years	0.369	−0.140	0.879	0.065	0.154
Smoking habits (presence)	8.432	−7.059	23.923	0.050	0.284
Regular exercise habits (presence)	−11.043	−26.179	4.093	−0.066	0.151
Frequency of vegetable consumption (every day)	−11.287	−24.515	1.941	−0.077	0.094
BMI	14.087	12.351	15.824	0.757	<0.001
Fasting blood glucose (g/L)	0.323	0.063	0.583	0.113	0.015
Present teeth (*n*)	−0.825	−2.114	0.463	0.652	0.208
BOP (presence)	7.812	−6.588	22.212	0.050	0.285
PPD score	8.432	−7.059	23.923	0.050	0.284
Model 2					
Gender (male)	1.380	−10.949	13.708	0.010	0.825
Age, years	0.384	−0.114	0.883	0.068	0.130
Smoking habits (presence)	6.951	−5.973	24.314	0.054	0.233
Regular exercise habits (presence)	−7.845	−22.806	7.116	−0.047	0.302
Frequency of vegetable consumption (every day)	−11.628	−24.557	1.302	−0.079	0.078
BMI	13.950	12.254	15.647	0.750	<0.001
Fasting blood glucose (g/L)	0.276	0.019	0.533	0.097	0.035
Present teeth (*n*)	−0.568	−1.823	0.687	−0.040	0.372
BOP (presence)	6.951	−5.973	24.314	0.054	0.233
CAL code	12.103	3.044	21.162	0.123	0.009

Adjusted R^2^ = 0.722 (Model 1) and 0.704 (Model 2). CI; confidence interval, VFA; visceral fat area, BMI; body mass index, BOP; bleeding on probing, PPD; probing pocket depth, CAL; clinical attachment level. Model 1; adjusted gender, age, smoking habits, regular exercise habits, frequency of vegetable consumption, BMI, fasting blood glucose, present teeth, BOP, and PPD score. Model 2; adjusted gender, age, smoking habits, regular exercise habits, frequency of vegetable consumption, BMI, fasting blood glucose, present teeth, BOP, and CAL code.

**Table 5 ijerph-20-01699-t005:** Factors associated with SFA in the study population by multiple regression analysis.

Dependent Variables	Unstandardized Coefficients (B)	95% CI for B		Standardized Coefficients Β	*p* Value
		Lower Bound	Upper Bound		
Model 1					
Gender (male)	−9.220	−25.356	6.917	−0.066	0.261
Age, years	−0.252	−0.908	0.404	−0.045	0.449
Smoking habits (presence)	−9.171	−29.096	10.755	−0.054	0.365
Regular exercise habits (presence)	−21.737	−41.206	−2.268	−0.130	0.029
Frequency of vegetable consumption (every day)	−15.637	−32.652	1.378	−0.106	0.071
BMI	13.107	10.874	15.341	0.705	<0.001
Fasting blood glucose (g/L)	−0.405	−0.740	−0.700	−0.142	0.018
Present teeth (*n*)	−0.402	−2.060	1.256	−0.029	0.633
BOP (presence)	11.586	−6.936	30.108	0.074	0.218
PPD score	0.498	−11.295	12.291	0.005	0.934
Model 2					
Gender (male)	−8.363	−24.528	7.801	−0.060	0.308
Age, years	−0.259	−0.912	3.950	−0.046	0.436
Smoking habits (presence)	−9.407	−29.262	10.449	−0.056	0.351
Regular exercise habits (presence)	−23.609	−43.226	−3.993	−0.142	0.019
Frequency of vegetable consumption (every day)	−15.561	−32.514	1.391	−0.106	0.072
BMI	13.204	10.980	15.428	0.710	<0.001
Fasting blood glucose (g/L)	−0.379	−0.715	−0.042	−0.133	0.028
Present teeth (*n)*	−0.581	−2.226	1.064	−0.041	0.486
BOP (presence)	12.455	−5.826	30.735	0.079	0.180
CAL code	−5.776	−17.654	6.102	−0.059	0.338

Adjusted R^2^ = 0.539 (Model 1) and 0.542 (Model 2). CI; confidence interval, SFA; subcutaneous fat area, BMI; body mass index, BOP; bleeding on probing, PPD; probing pocket depth, CAL; clinical attachment level. Model 1; adjusted gender, age, smoking habits, regular exercise habits, frequency of vegetable consumption, BMI, fasting blood glucose, present teeth, BOP, and PPD score. Model 2; adjusted gender, age, smoking habits, regular exercise habits, frequency of vegetable consumption, BMI, fasting blood glucose, present teeth, BOP, and CAL code.

## Data Availability

The data presented in this study are available on request from the corresponding author. The data are not publicly available due to ethical restrictions.

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
