# Peer review of "Association between Periodontal Condition and Fat Distribution in Japanese Adults: A Cross-Sectional Study Using Check-Up Data"

_ijerph, 2023, doi:10.3390/ijerph20031699_

Round 1
Reviewer 1 Report
Very interesting article on a very pertinent topic. Overall the design is sufficient. Minor English edits are required. The authors should explain whether the Questionnaire they used to assess lifestyle was their own creation. Was the Questionnaire piloted? The conclusion needs to be broadened and maybe a section should be dedicated to future direction of research. Dietary guidelines could be an interesting avenue of future research.
The authors need to connect their research findings with implications for the future. Methodology is fine. English grammar and synthex needs improvement throughout.
Author Response
Our comments: Thank you for your professional comment. As for the questionnaire, we referred to previous literature (Iwasaki et al., 2019). We have added the reference (Line 108).
In addition, we have added the sentence in the conclusion section about future direction of research (Line 255-256). As you pointed, we need to consider useful ways to reduce visceral fat through diet and exercise.
As for English grammar, we have checked by a native English speaker.
Thank you again, for your nice suggestions.
Reviewer 2 Report
Lot of studies in relation to visceral fat content and Obesity in relation to BMI has been published so far. It would have been better to add any biomarker in relation to fat content (visceral and subcutaneous) for appropriate results.
Author Response
Our comments: Thank you for your nice comment. We agree to your comment, and it would be better to add biomarker in relation to fat content. Unfortunately, this study was based on health checkups data, we would like to make use of it in our next research.
We added this issue as the limitation (Line 249-251).
Thank you again, for your nice suggestions.
Reviewer 3 Report
The manuscript titled “ Association between Periodontal Condition and Fat Distribution in Japanese Adults: A Cross-sectional Study Using Check-up Data” is an interesting study which is well written. I have very few suggestions related to the manuscript.
In the introduction part page 1 line 39 it’s written as excessive fact - it must be fat
In the discussion while interpreting the results of regression analysis there was no importance given to the regression coefficients. The correlation coefficient between VFA and CAL was only 0.123 eventhough it was statistically significant. It indicate significant but weak correlation. Whereas VFA with BMI was 0.7 indicating a strong correlation. It would be appropriate to indicate the weaker correlation while interpreting your results
Author Response
Our comments: Thank you for your professional comment. We have revised the words, fact to fat (Line 39).
We agree to your suggestions, we have added the limitation about weak correlation for VFA and CAL in discussion sections (Line 240-242).
Reviewer 4 Report
-A lot of the introduction literature is only limited to specific groups, however they are strong/er papers from other international groups. Please consider revision of refferenced studies.
-Although you build up the argument that VFA, SFA are better descriptors of obesity than BMI, your results indicate a strong correlation of BMI with VFA and SFA. However, no comments are made on this in the discussion. Please comment on this. Furthermore, did you do a BMI and PPD and CAL analyses?
Author Response
Our comments: Thank you for your professional comment. We have added the new referenced.
As for BMI, we investigated the correlation analysis between BMI with PPD and CAL. Interestingly, VFA values (PPD; r= 0.251, p=0.001, CAL; r=0.353, p<0.001) were more positive correlated with PPD and CAL, compared to BMI values (PPD; r= 0.197, p=0.013, CAL; r=0.227, p=0.004). We added the comments about it in discussion section (Line 243-246).
Round 2
Reviewer 4 Report
Lines 250-252 : A double negative in the sentence. Does this need revision?